# Algorithmic voice transformations reveal the phonological basis of language-familiarity effects in cross-cultural emotion judgments

Tomoya Nakai [1,2☯]*, Laura Rachman [3☯], Pablo Arias Sarah [4,5,6], Kazuo Okanoya [7,8], Jean-Julien Aucouturier [5,9]*

**1** Lyon Neuroscience Research Center (CRNL), (INSERM/CNRS/University of Lyon), Bron, France, **2** Center for Information and Neural Networks, National Institute of Information and Communications Technology, Suita, Japan, **3** Department of Otorhinolaryngology, University Medical Center Groningen, University of Groningen, Groningen, The Netherlands, **4** Lund University Cognitive Science, Lund University, Lund, Sweden, **5** Sciences et Technologies de la Musique et du Son (IRCAM/CNRS/Sorbonne Université), Paris, France, **6** School of Psychology & Neuroscience, University of Glasgow, Glasgow, United Kingdom, **7** The University of Tokyo, Graduate School of Arts and Sciences, Tokyo, Japan, **8** Advanced Comprehensive Research Organization, Teikyo University, Tokyo, Japan, **9** FEMTO-ST Institute (CNRS/Université de Bourgogne Franche Comté), Besançon, France

☯ These authors contributed equally to this work.
* nakai.tomoya@neuro.mimoza.jp (TN); aucouturier@gmail.com (JJA)

**Data Availability Statement:** All analysis codes and data are available from Open Science Framework (https://osf.io/4py25/).

## Abstract

People have a well-described advantage in identifying individuals and emotions in their own culture, a phenomenon also known as the other-race and language-familiarity effect. However, it is unclear whether native-language advantages arise from genuinely enhanced capacities to extract relevant cues in familiar speech or, more simply, from cultural differences in emotional expressions. Here, to rule out production differences, we use algorithmic voice transformations to create French and Japanese stimulus pairs that differed by exactly the same acoustical characteristics. In two cross-cultural experiments, participants performed better in their native language when categorizing vocal emotional cues and detecting non-emotional pitch changes. This advantage persisted over three types of stimulus degradation (jabberwocky, shuffled and reversed sentences), which disturbed semantics, syntax, and supra-segmental patterns, respectively. These results provide evidence that production differences are not the sole drivers of the language-familiarity effect in cross-cultural emotion perception. Listeners' unfamiliarity with the phonology of another language, rather than with its syntax or semantics, impairs the detection of pitch prosodic cues and, in turn, the recognition of expressive prosody.

## Introduction

In everyday life, we interact socially with a variety of other people, some of them from cultural groups with whom we have had limited previous encounters. A large body of psychological evidence shows that familiarity, or lack thereof, with the cultures of others crucially affects how we process their social signals, such as their facial or vocal expressions. People recognize

**Funding:** This work was supported by ERC Grant StG 335536 CREAM, Fondation pour l'Audition FPA RD-2018-2, ANR REFLETS and SEPIA to J.J.A., JSPS KAKENHI (20H05023 and 20K07718) and H2020 Marie Skłodowska-Curie Actions (101023033) to T.N., JSPS KAKENHI Grant-in-Aid for Scientific Research on Innovative Areas (#4903; Evolinguistics) to K.O. The funders had no role in study design, data collection and analysis, decision to publish, or preparation of the manuscript.

**Competing interests:** The authors have declared that no competing interests exist.

faces of their own race more accurately than faces of other races (other-race effect [1, 2]), and identify speakers of their native language better than speakers of other languages (language-familiarity effect or LFE; [3, 4]). Even within a given cultural or language group, familiarity with a given speaker's voice facilitates how their spoken words are remembered [5] or how prosodic cues are disambiguated [6], although recent evidence suggests it does not improve the consistency of judgements of speaker traits [7]. These familiarity effects are thought to result primarily from perceptual learning; differential exposure warps an observer's perceptual space to facilitate discrimination of common in-group items. Comparatively rare out-group items are thus encoded less efficiently [8, 9]. Such perceptual warping is manifest in the fact that listeners rate pairs of speakers of their own language as more dissimilar than pairs of speakers of the other language [4], or that young infants can discriminate sounds from all languages equally well before six months of age, but develop a native-language advantage by about 6-12mo [10, 11].

One cognitive domain in which language-familiarity effects may be particularly prevalent is that of cross-cultural inferences of emotional prosody [12–15]. Judging whether a particular speech utterance is unusually high or low, a given phoneme bright or dark, whether a specific pitch inflection is expressive or phonological would all appear to be advantaged, or conversely impaired, by acquired auditory representations optimized for the sounds of one language or another [16, 17]. Most cross-cultural data indeed reveal an in-group advantage for identifying emotions displayed by members of the same rather than a different culture (see [18, 19] for a review).

However, it has been difficult to conclusively determine to what extent such differences arise from perceptual learning effects in the sense of the above or, perhaps more simply, from cultural differences in how these emotions are expressed [18]. For instance, perceptual learning would predict better cross-cultural recognition with increasing language similarity, but such evidence is mixed. Scherer and colleagues found Dutch listeners better at decoding German utterances than listeners of other, less similar European and Asian languages [12]. However, other studies found no differences in, for example, how Spanish listeners identified vocal emotions from the related English and German languages, and the unrelated Arabic [14], or how English listeners processed utterances in the related German language and the unrelated Tagalog of the Philippines [13]. Most critically, because most of such cross-cultural investigations use stimuli recorded by actors of each language, differences in the agreement levels across groups may simply arise because of cultural differences in emotional expressions. If the last author, a Frenchman, has difficulties processing emotional cues spoken by the first author, is it because his auditory representations of the Japanese phonetic inventory are poorly differentiated (see for example, [20]), or because one simply does not use the same cues to express *joy* in Japanese and in French (see for example, [21])? To progress on this debate, it would be necessary to employ emotional voice stimuli which, while being recognized as culturally-appropriate emotional expressions in different languages, utilize *exactly* the same prosodic cues in *exactly* the same manner (for example, a 50-cent pitch increase on the second syllable), something which is at best impractical with vocal actors.

To this aim, we used audio processing software called DAVID [22] to apply a fixed set of programmable acoustical/emotional transformations to pre-recorded voice samples in two languages (French and Japanese), and performed cross-cultural voice recognition experiments to assess how identical cues were processed by speakers of both languages. In brief, DAVID alters the acoustic features of original voices by combining audio effects of pitch shift, vibrato, inflection, and filtering, to the result of modifying the emotional content of the original voice (**Fig 1**). In two previous studies with DAVID, authors applied real-time emotion transformations on participants' voices while they were talking. Transformed voices were perceived as

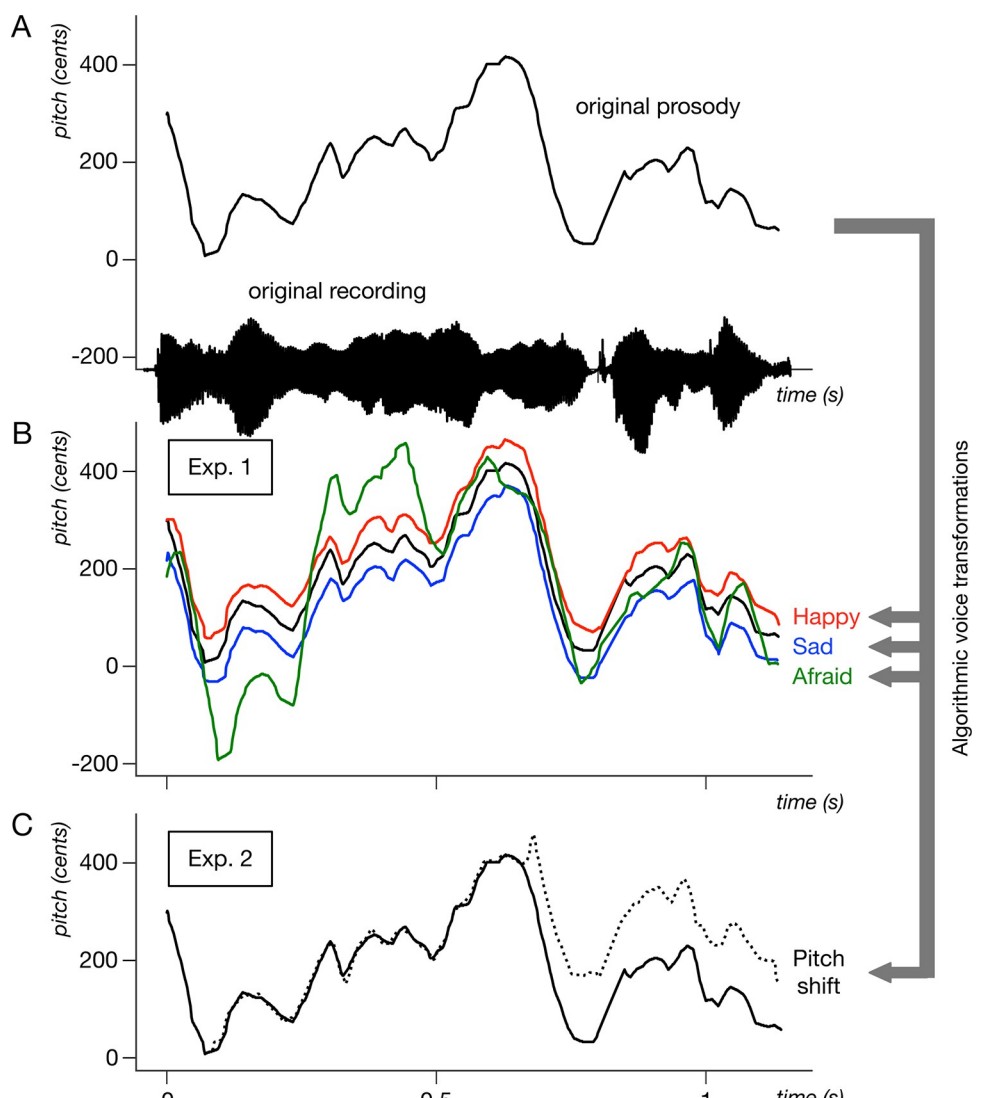

**Fig 1. Illustration of the algorithmic voice transformations used in the study.** A single recording of a French female speaker saying "*J'ai oublié mon pardessus*" (I forgot my jacket) is manipulated with the DAVID voice transformation platform to make it sound more happy, sad, or afraid (Experiment 1) or to insert a sudden pitch shift in the middle of the sentence (Experiment 2). (**A**) Solid black: time series of pitch values in the original recording estimated with the SWIPE algorithm [25]. The speech waveform of the unmodified recording is shown on the x-axis. Pitch values on y-axis are normalized to cents with respect to mean frequency 200 Hz. (**B**) Red, blue, and green lines: pitch of manipulated audio output in the Happy, Sad, and Afraid transformations, respectively, as used in Experiment 1. (**C**) Dashed line: +150 cents pitch shift occurring at *t* = 700 ms, Experiment 2.

natural emotional expressions, to the point that most participants remained unaware of the manipulation, but they nevertheless affected participants' emotional states and skin conductance responses [23, 24]. DAVID was validated in multiple languages, finding that transformed emotions were recognized at above-chance levels when applied to either French, English, Swedish, or Japanese utterances, and with a naturalness comparable to authentic speech [22].

In a series of two experiments, we manipulate here both Japanese (JP) and French (FR) voices with the same set of parametric transformations by DAVID, so as to display emotions of happiness, sadness, and afraid (Experiment 1) or to create sudden pitch shifts in the middle

of a sentence without explicit emotional meaning (Experiment 2). Importantly, this procedure excludes the possible effect of cultural variability in how these cues were expressed, because target categories in the two languages were produced exactly in the same manner (i.e., with the same algorithmic parameters). We then let two groups of native JP and FR speakers listen to pairs of voices (one neutral, one manipulated), in both languages, and either categorize the emotion of the second voice (Experiment 1) or detect which of the two extracts had a pitch change (Experiment 2). To test which linguistic component contributed to an LFE, if any, we further presented the same participants with syntactically/semantically modified sentences (see Materials and Methods for the detailed explanation), as well as time-reversed voices and sentences in an unfamiliar language (Swedish, SE).

## Materials and methods

### Ethics

All experiments were approved by the Institut Européen d'Administration des Affaires (INSEAD) IRB, National Institute of Information and Communications Technology, and the departmental review board of the University of Tokyo, Komaba. All participants gave their written informed consent and were debriefed and informed about the true purpose of the research immediately after the experiment.

### Recordings and experimental apparatus

We prepared four normal JP sentences and four normal FR sentences, translated from the same four semantically-neutral English sentences taken from Russ et al. (2008) [26]. Jabberwocky variants [27] of the eight sentences were created by replacing each content word in the normal sentences with a pseudo-word with the same number of syllables, without changing functional words (for example, JP: *Uwagi wo wasureta* (I forgot my jacket) -> *Etaki\* wo morushita\**). Jabberwocky sentences did not have any meaning (or content), but still maintained the grammatical structures of their original languages. Shuffled Jabberwocky sentences were produced by changing the word order of the corresponding Jabberwocky sentences so that they did not maintain any grammatical structures of their original languages (for example, JP: *Etaki wo morushita -> Wo\* morushita\* etaki\**; see **Table 1**). The number of syllables was balanced across the two languages (7.25 ± 1.5 [mean ± SD] for the normal JP stimuli, and 7.5 ± 1.3 for the normal FR sentences), and the Jabberwocky and shuffled variants of each sentence had the same number of syllables as the original version, in both languages.

A male and female JP native speaker then recorded the four normal JP sentences, four Jabberwocky JP sentences, and four shuffled JP sentences. We used the four utterances of the normal JP sentences to make four reverse JP recordings by playing them backwards. Similarly, we recorded four normal FR sentences, four Jabberwocky FR sentences, four shuffled FR sentences, and produced four reverse FR sentences in the same manner, with a male and female FR native speaker. The recordings took place in a sound-attenuated booth, using Garage-Band software (Apple Inc.), a headset microphone (DPA 4066), and an external soundcard (Fireface UCX). In addition, we used four male and four female normal Swedish (SE) recordings from a previous study [22]. Although these SE sentences did not contain the same semantic information as JP and FR sentences, we considered them as a baseline control for unfamiliar language for both participant groups. All stimuli had a duration of 1.5s and were normalized in root mean square intensity.

**Table 1. French and Japanese normal, jabberwocky, and shuffled sentences used in this study.** English translations are added for clarification but were not included in this study.

| JP sentences | FR sentences | English translations |
|---|---|---|
| Normal | | |
| Uwagi wo wasureta | J'ai oublié mon pardessus | I forgot my jacket |
| Kaigi ni itta | Je suis allé à la réunion | I attended the meeting |
| Hikouki wa ippai | L'avion est complet | The airplane is full |
| Hyoumen wa nameraka | La surface est lisse | The surface is slick |
| Jabberwocky | | |
| Etaki wo morushita | J'ai odrié mon tarpodu | |
| Goito ni etta | Je suis ijé à la boussion | |
| Komeubi wa ottai | L'ation est bondret | |
| Bouren wa soniyaka | La borpaque est nitte | |
| Shuffled | | |
| Wo morushita etaki | Odrié j'ai tarpodu mon | |
| Etta ni goite | Boussion ijé la je à suis | |
| Komeubi ottai wa | L'est ation bondret | |
| Bouren soniyaka wa | La est nitte borpaque | |

## Audio manipulation algorithm

The audio recordings were manipulated with the software platform DAVID [22], available at https://forum.ircam.fr/projects/detail/david/. For the current experiment, we used predetermined pitch shift, vibrato, inflection, and filtering transformations designed to evoke happy, sad, and afraid expressions. **Table 2** describes the transformation parameter values used in this study, which were applied identically to both Japanese and French-language stimuli. These values were based on previous studies [22, 23] that validated that they sounded both natural and recognizable in both JP and FR. Note that we modified some of the parameters from Rachman et al. [22]; the pitch shift in the sad voices was set to −50 cents to mirror that in the happy voices. We also applied the same vibrato parameters to male and female voices because we considered that the original vibrato parameters would not have enough effect on some of the female JP voices. Examples of the manipulations are illustrated in **Fig 1**, and in supplementary audio files (**S1 File**).

**Table 2. Parameter values of the three emotional transformations used in Experiment 1 (for details, refer to main text and [22]).**

| | Transformations | | |
|---|---|---|---|
| | Happy | Sad | Afraid |
| **Pitch** | | | |
| shift, *cents* | +50 | −50 | – |
| **Vibrato** | | | |
| rate, *Hz* | – | – | 8.5 |
| depth, *cents* | – | – | 30 |
| **Inflection** | | | |
| duration, *ms* | 500 | – | 150 |
| min., *cents* | −200 | – | −200 |
| max., *cents* | +140 | – | +200 |
| **Filter** | | | |
| cut-off, *Hz* | > 8000 | < 8000 | |
| Slope, *dB/octave* | + 9.5 | −12 | |

Pitch shifting denotes the multiplication of the fundamental frequency of the original voice signal by a constant factor $\alpha$ (for example, + 50 cents, a 2.93% change of F0). Vibrato is a periodic modulation of the fundamental frequency, implemented as a sinusoidal modulation of the pitch shift effect, with a rate parameter (for example, 8 Hz), a depth (for example, + 40 cents) and a random variation of the rate (for example, 30% of the rate frequency). Inflection is a rapid modification of the fundamental frequency at the start of each utterance, which overshoots its target by several semitones but quickly decays to the normal value. Finally, filtering emphasizes or attenuates the energy of certain areas of the frequency spectrum (for example, a high-shelf filter with a cut-off frequency at 8000 Hz, +9.5 dB per octave). Full details of these algorithms can be found in [22].

## Exp. 1—Emotion recognition

### Participants

A total of 44 participants were recruited in Experiment 1. The participants consisted of 22 native Japanese (JP) speakers (9 female, M = 19.7) and 22 French (FR) speakers (12 female, M = 23.6). None of the JP speakers had ever learned French, and none of the FR speakers had ever learned Japanese. Experiments with the Japanese speakers were conducted in the University of Tokyo (Japan), while those with the French speakers were conducted at the INSEAD/Sorbonne-Université Behavioural platform in Paris (France). Volunteers were recruited through local databases and mailing lists and were financially compensated for their participation. Two French participants were excluded because they did not satisfy language requirements (not native FR speakers). Furthermore, one Japanese participant was excluded because they reported that they could hear the auditory stimuli only from one side of the headphone.

### Stimuli

All sentence recordings described above (incl. reverse JP, reverse FR, and normal SE) were processed with DAVID and transformed into happy, sad, and afraid variants, resulting in 96 manipulated recordings for both JP and FR (3 emotions × 4 sentences × 4 conditions × 2 speakers), and 24 manipulated SE recordings.

### Procedure

In each trial, participants listened to pairs of utterances of the same sentence by the same speaker (presented with an inter-stimuli interval = 0.7–1.3 s). The first utterance was always the neutral recording and the second utterance was either the same recording unprocessed (neutral condition) or processed with one of the emotional transformations (happy, sad, afraid). After hearing the two utterances, four emotion labels with corresponding keys were presented visually ("[S] happy", "[D] sad", "[F] afraid", "[G] identical"); participants (all right-handed) were instructed to answer whether the second utterance sounded happy, sad, afraid, or neutral by pressing one of four keys ("S", "D", "F", and "G") with their left fourth, third, second, and first finger, respectively. The next trial started when participants pressed the "ENTER" key with their right first finger. All participants were presented with the 96 JP pairs, 96 FR pairs, and 24 SE pairs, randomized across participants in one single block. The correspondence of keys and response categories was randomized across trials. Visual stimuli were displayed on a laptop-computer screen, and the voice stimuli were presented through closed headphones. Stimulus presentation and data collection were controlled using PsychoPy toolbox [28]. Response categories in the recognition task for the French group in fact used the English terms (happy, sad, afraid) instead of the French equivalents, but were defined in the

instructions using the equivalent French terms. We have confirmed in our previous study that emotion recognition tasks can be performed without problems in this presentation setting [22]. Response categories used in Japanese groups were presented in Japanese terms.

## Data analysis

We computed the mean hit rate (biased hit rate, $H_b$) of the three emotion categories (happy, sad, and afraid) for each of the nine conditions (JP, FR: normal, jabberwocky, shuffled, reversed; SE: normal). To take a possible response bias into account, we also calculated unbiased hit rates ($H_u$) and chance proportions ($p_c$) for each participant [29]. Unbiased hit rates take a value between zero and one and take into account how often an emotion is identified correctly, as well as the total number of times that an emotion label is used. Both biased and unbiased hit rates are reported in the Results section, but the unbiased hit rates were submitted to paired *t*-tests to assess performance against chance level and to mixed ANOVAs to test for effects of language and sentence type. Alpha level was set at .05 and significant effects were followed up by paired *t*-tests using Bonferroni correction for multiple comparisons. Effect sizes are shown as Cohen's *d* or partial eta squared ($\eta_p^2$). Finally, for easier comparison with other emotion recognition studies using different numbers of response categories, we also report the proportion index (*pi*). *pi* expresses the biased hit rate transformed to a standardized scale ranging from 0 to 1, where a score of 0.5 indicates chance performance and a score of 1 indicates a decoding accuracy of 100% [30].

## Exp. 2—Pitch change detection

### Participants

A Total of 45 participants were recruited in Experiment 2. The participants consisted of 24 native French (FR) speakers (12 female, 21 right-handed, mean age = 22.3, SD = 2.9 years) and 21 native Japanese (JP) speakers (6 female, all right-handed, mean age = 23.0, SD = 3.7 years) participated in this study. One JP male participant was excluded from the analyses because he reported to have changed strategies between the calibration and the test phase, resulting in 100% accuracy performance across all conditions. Experiments with the Japanese speakers were conducted at the Center for Information and Neural Networks (Japan), and those with the French speakers were conducted at the INSEAD/Sorbonne-Université Behavioural platform in Paris (France).

### Stimuli

Stimuli were based on the 64 non-manipulated French and Japanese sentences used in Experiment 1 (2 languages × 4 sentences × 4 conditions × 2 speakers). We used the pitch-shift subcomponent of DAVID to create sudden positive pitch changes at a random time point between 400 and 600 ms after sentence onset, and asked participants whether they could detect their occurrence. The size of the shift, between +14 and +257 cents, was calibrated with an adaptive procedure (see Procedure below).

### Procedure

Prior to the test phase, an adaptive staircase procedure was used to determine participants' pitch detection threshold and equalize native-language performance among participants. The adaptive procedure involved a two-interval forced choice (2IFC) paradigm with a one-up, two-down progression rule to converge to a 70.7% performance level on the psychometric function [31, 32]. In each trial, one of the four normal-type sentences in the participant's native

language (**Table 1**) was randomly selected and presented twice. A pitch shift was applied at a random point in one of the two sentences and participants were asked to respond in which sentence they heard the pitch change. Responses were made with the left middle and index fingers by pressing one of two buttons on a keyboard. The initial pitch shift was set at 200 cents and the initial step size was fixed at 40 cents and halved after every two reversals until a 5 cents step size that was maintained until the end of the procedure. The onset of the pitch shift in each trial was randomly set at 500 ± 100 ms into the recordings. After twelve reversals, the adaptive procedure was terminated and the average of the pitch shift values of the last six reversals was taken to estimate the value targeting a 70.7% response accuracy. Participants performed two separate adaptive staircase procedures for male and female speakers, the order of which was counterbalanced across participants.

In the test phase, participants then performed the same 2IFC task, this time using the fixed pitch shift magnitude determined in the adaptive procedure. The same pitch shift (for speakers of the corresponding sex) was applied to all stimuli, regardless of language or stimulus degradation. Participants took part in three blocks of 64 trials. In each block, they were presented with pairs of neutral and manipulated versions of each of the 64 recordings (4 sentences × 4 conditions × 2 speakers × 2 languages) and had to indicate in which of the stimuli of each pair they heard the pitch change. The 64 sentences were randomized within each block and the onset of the pitch shift was randomized across all 192 trials.

## Data analysis

We computed the accuracy as the percentage of correct responses for each of the eight conditions (FR, JP: normal, jabberwocky, shuffled, reversed). The accuracy and response time data were then submitted to mixed ANOVAs to test for language effects. Significant effects were followed up by *t*-tests using Bonferroni corrections for multiple comparisons and effect sizes are shown as Cohen's *d* or partial eta squared ($\eta_p^2$). Mixed three-way ANOVA of Language × Participant group × Condition on the response times did not show any significant main effect or interaction (all *p*s > .05), and are not reported in main text.

## Data and code availability

The source data and analysis code used in the current study are available from Open Science Framework (https://osf.io/4py25/ [33]).

## Results

### Experiment 1: Emotion categorization

We analyzed the results of 41 participants (N = 20 for FR participants and N = 21 for JP participants) who listened to both FR, JP and SE-language stimuli and categorized the emotions by the voice transformations, which were identical in both languages (**Fig 1B**).

Both participant groups recognized the three emotional transformations above chance-level when applied to normal speech of their native language (see **Table 3**, Bonferroni corrected across twelve comparisons). For FR participants, unbiased hit rates of the three emotion categories and neutral condition were significantly larger than their corresponding chance proportions in normal FR sentences (*p* < .01). For JP participants, unbiased hit rates were also larger than chance for all three emotions and neutral condition in both normal JP and normal FR sentences (*p* < .01). Participants' performance (Happy: native FR *pi* = .66, native JP *pi* = .80; Sad: native FR *pi* = .68, native JP *pi* = .64, Afraid: native FR pi = .58, native JP *pi* = .65; *pi*: proportion index, see Materials and Methods.) was comparable with a previous study using

**Table 3.** FR = French; JP = Japanese; SE = Swedish; $H_b$ = biased hit rate (%); $pi$ = proportion index; $H_u$ = unbiased hit rate (%); $p_c$ = chance proportion (%); $t$ = t-score, degrees of freedom are 19 and 20 for the FR and JP groups respectively.

| | | | Biased | | Unbiased | | |
| --- | --- | --- | --- | --- | --- | --- | --- |
| **Group** | **Stimuli** | | $H_b$ | $pi$ | $H_u$ | $p_c$ | $t$ |
| FR | FR | Happy | 39.5 | 66.2 | 31.8 | 3.5 | 5.4*** |
| | | Sad | 41.2 | 67.8 | 21.6 | 6.0 | 3.4** |
| | | Afraid | 32.2 | 58.7 | 18.8 | 4.5 | 3.5** |
| | | Neutral | 86.0 | 94.9 | 45.8 | 11.0 | 8.1*** |
| | | **Mean** | 49.7 | 74.8 | | | |
| | JP | Happy | 29.2 | 55.3 | 21.3 | 3.3 | 4.0** |
| | | Sad | 20.9 | 44.3 | 13.9 | 2.6 | 2.7 |
| | | Afraid | 13.5 | 32.0 | 7.7 | 2.0 | 3.2* |
| | | Neutral | 91.0 | 96.8 | 31.8 | 1.7 | 5.7*** |
| | | **Mean** | 38.7 | 65.4 | | | |
| | SE | Happy | 27.2 | 52.8 | 18.3 | 3.2 | 3.7** |
| | | Sad | 25.9 | 51.1 | 19.3 | 3.0 | 3.0* |
| | | Afraid | 17.9 | 39.5 | 9.9 | 2.8 | 3.1* |
| | | Neutral | 88.2 | 95.7 | 31.1 | 15.7 | 5.2*** |
| | | **Mean** | 40.0 | 66.5 | | | |
| JP | FR | Happy | 33.5 | 60.2 | 27.9 | 2.7 | 7.2*** |
| | | Sad | 64.7 | 84.6 | 33.0 | 8.9 | 5.6*** |
| | | Afraid | 25.2 | 50.2 | 16.9 | 2.9 | 3.9** |
| | | Neutral | 87.0 | 95.3 | 48.9 | 10.8 | 10.8*** |
| | | **Mean** | 52.6 | 76.9 | | | |
| | JP | Happy | 57.1 | 79.9 | 49.2 | 4.3 | 7.2*** |
| | | Sad | 37.7 | 64.5 | 23.0 | 4.6 | 4.1** |
| | | Afraid | 38.7 | 65.4 | 26.8 | 4.2 | 5.7*** |
| | | Neutral | 89.1 | 96.1 | 42.5 | 11.6 | 11.5*** |
| | | **Mean** | 55.6 | 79.0 | | | |
| | SE | Happy | 36.1 | 62.8 | 26.4 | 3.3 | 8.7*** |
| | | Sad | 25.4 | 50.6 | 14.2 | 4.0 | 2.4 |
| | | Afraid | 15.1 | 34.7 | 10.2 | 1.9 | 2.3 |
| | | Neutral | 90.1 | 96.4 | 34.8 | 1.6 | 8.7*** |
| | | **Mean** | 41.6 | 68.2 | | | |

*$p < .05$

**$p < .01$

***$p < .001$, Bonferroni corrected across twelve comparisons.

DAVID (Happy: $pi$ = .66, Sad: $pi$ = .71, Afraid: $pi$ = .70) [22]. Together these results show that participants had good emotion recognition abilities in their native language and that the emotional transformations are appropriate in both languages.

To test for language-familiarity effects, we then examined unbiased hit rates on cross-cultural stimuli. In this analysis, we averaged unbiased hit rates in the three emotional categories (happy, sad, afraid). The neutral condition was not included in this analysis because this condition did not contain any emotional transformation. A mixed analysis of variance (ANOVA) with Language as within-subject factor and Participant group as between-subject factor showed a significant interaction ($F(1,39) = 19.38$, $p < .001$, $\eta_p^2 = 0.33$), as well as a significant main effect of participant group ($F(1,39) = 7.35$, $p = .010$, $\eta_p^2 = 0.16$). The post-hoc $t$-tests

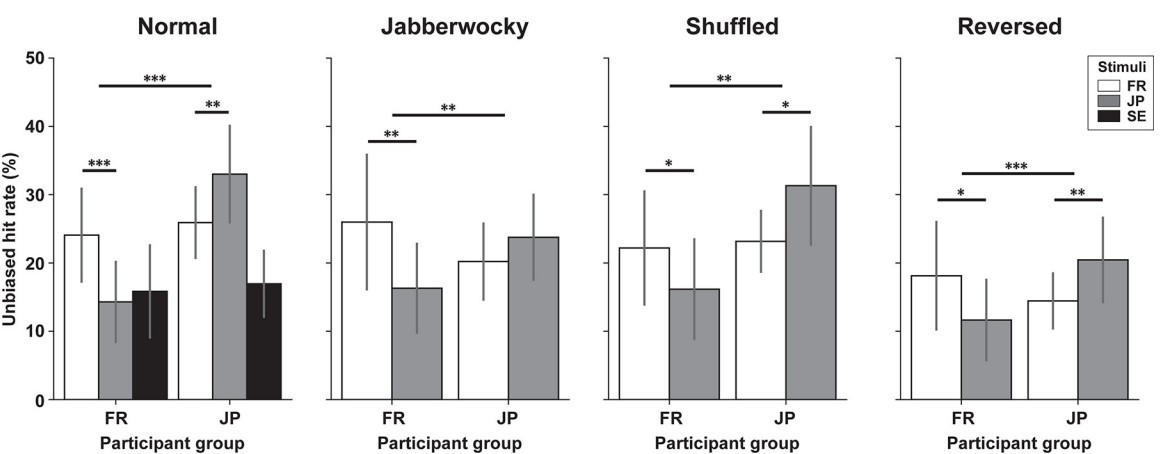

**Fig 2. Emotion categorization: Unbiased hit rates averaged over the three non-neutral emotion categories, grouped by normal (French/FR, Japanese/JP and Swedish/SE stimuli), jabberwocky (FR, JP), shuffled (FR, JP), and reversed speech (FR, JP) conditions, for two groups of FR (N = 20) and JP (N = 21) native speakers.** *p < .05. **p < .01, ***p < .001, Bonferroni adjusted. Error bars, 95% CI.

($\alpha_{Bonfer}$ = 0.025) showed that FR participants could detect emotion categories of normal FR sentences significantly better than in normal JP sentences ($t(19)$ = 3.50, $p$ = .001, $d$ = 0.72), and JP participants could detect emotions in normal JP sentences significantly better than in normal FR sentences ($t(20)$ = 2.70, $p$ = .007, $d$ = 0.52; see **Fig 2**). In addition, both participants groups detected their native language stimuli better than SE stimuli ($t(19)$ = 2.73, $p$ = .007, $d$ = 0.57; $t(20)$ = 4.26, $p$ < .001, $d$ = 1.21). These results indicate a clear LFE on the recognition of emotional prosody—importantly, even as the cues in both languages were computer-manipulated to be identical.

This LFE was observed in a quasi-identical manner in all three degraded FR and JP stimulus conditions, although these manipulations degraded some of the recognition to chance-level. Unbiased hit rates showed a significant interaction of Language × Participant group in Jabberwocky ($F(1,39)$ = 9.43, $p$ = .004, $\eta_p^2$ = 0.19), shuffled ($F(1,39)$ = 8.89, $p$ = .005, $\eta_p^2$ = 0.19), and reversed sentences ($F(1,39)$ = 13.40, $p$ < .001, $\eta_p^2$ = 0.26). FR participants were more accurate to attribute vocal cues to emotions in Jabberwocky FR than Jabberwocky JP ($t(19)$ = 2.78, $p$ = .006, $\alpha_{Bonfer}$ = 0.025, $d$ = 0.54; **Fig 2**), shuffled FR than shuffled JP ($t(19)$ = 2.20, $p$ = .020, $d$ = 0.36), and in reverse FR than reverse JP ($t(19)$ = 2.41, $p$ = .013, $d$ = 0.44). JP participants were more accurate in shuffled JP than shuffled FR ($t(20)$ = 2.12, $p$ = .023, $d$ = 0.54), in reverse JP than reverse FR ($t(20)$ = 2.82, $p$ = .005, $d$ = 0.52; **Fig 2**). While not significant at the corrected $\alpha_{Bonfer}$ = 0.025 level, response patterns of JP participants in Jabberwocky JP or FR sentences ($t(20)$ = 1.38, $p$ = .092, $d$ = 0.27) were also in the same direction.

To examine whether the LFE depends on the different emotional categories, we also averaged unbiased hit rates in the four sentence types (normal, reverse, jabberwocky, shuffled) and compared across the three emotional categories (happy, sad, afraid; **Fig 3**). For FR participants, the LFE affected all three emotions identically. A 2-way repeated measures ANOVA of Emotion category × Language type showed significant main effects of Emotion ($F(2,38)$ = 4.06, $p$ = .025, $\eta_p^2$ = 0.18) and Language ($F(1,19)$ = 31.01, $p$ < .001, $\eta_p^2$ = 0.62), but no interaction. For JP participants, we found both significant main effects of Emotion ($F(2,40)$ = 8.83, $p$ < .001, $\eta_p^2$ = 0.31), Language ($F(1,20)$ = 13.56, $p$ = .002, $\eta_p^2$ = 0.40) and an interaction ($F(2,40)$ = 11.02, $p$ < .001, $\eta_p^2$ = 0.36), reflecting LFE-like effects of Language for Happy and Afraid emotions (both $p$ < .001 with post-hoc $t$-tests), but not Sad ($p$ = .90).

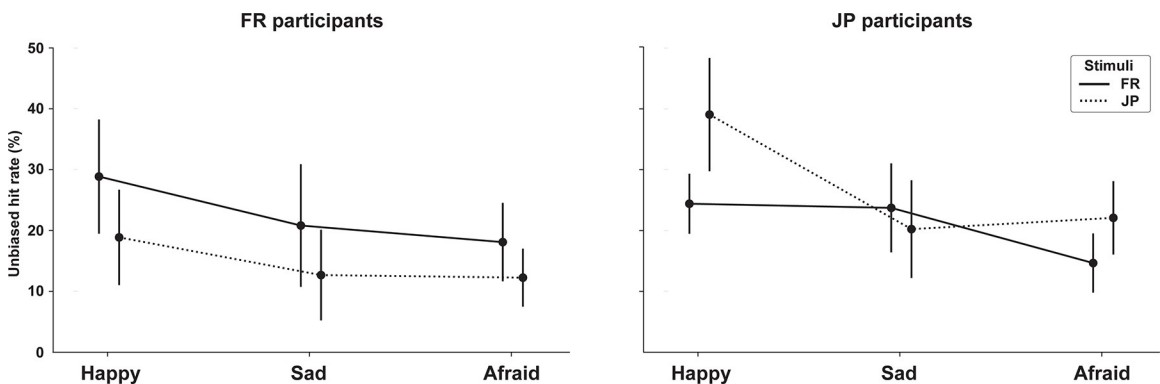

**Fig 3.** Unbiased hit rates for each emotion category for both FR (Left) and JP participants (Right), averaged across normal, reversed, jabberwocky, and shuffled conditions. Solid line shows unbiased hit rat of FR sentences, while dashed line shows that of JP sentences. Error bars, 95% CI.

### Experiment 2: Pitch shift detection

In Experiment 1, interactions of language and participant groups in emotion recognition were preserved even when higher-level characteristics of speech such as semantics, syntax or supra-segmental patterns were degraded. This suggests that the effect of language familiarity is based on non-emotional and non-cultural differences in auditory capacities such as pitch processing. In a second experiment, we therefore tested two new groups of FR (N = 24) and JP (N = 20) participants on their ability to detect sudden pitch changes (**Fig 1C**) in full sentences of their native and foreign languages. In addition, to address the different accuracy in the two partici-pant groups, Experiment 2 used an adaptive procedure to calibrate participants' performance at a standard level.

The average pitch shift calibrated for FR participants was M = +70 cents, and M = +81 cents for JP participants. The pitch shift values were submitted to a mixed 2 × 2 (Participant group × Speaker sex) ANOVA, which revealed a significant main effect of Speaker sex ($F(1,42)$ = 7.93, $p$ = .007, $\eta_p^2$ = 0.16): pitch shifts applied to female voices were larger than for male voices (+85 vs. +64 cents, $t(43)$ = 2.77, $p$ = .004, $d$ = 0.47). There was no main effect of Partici-pant group, and no interaction between Participant group and Speaker sex.

A mixed 2 × 2 × 4 (Participant group × Language × Condition) ANOVA was conducted on accuracy scores. There was a main effect of Condition ($F(3,126)$ = 5.96, $p$ < .001, $\eta_p^2$ = 0.12), in which pitch shifts, irrespective of language, were more accurately detected in normal sen-tences than in shuffled ($t(43)$ = 3.49, $p$ < .001, $\alpha_{Bonfer}$ = 0.05/6 = .0083, $d$ = 0.40) and reversed sentences ($t(43)$ = 3.40, $p$ < .001, $d$ = 0.40; no differences between any of the other sentence types). More importantly, as in Experiment 1, there was a significant interaction between Par-ticipant group and Language ($F(1,42)$ = 21.05, $p$ < .001, $\eta_p^2$ = 0.33), indicating a clear LFE on the detection of pitch shifts: accuracy in the FR participant group was higher for FR stimuli than for JP stimuli ($t(23)$ = 3.83, $p$ < .001, $\alpha_{Bonfer}$ = .025, $d$ = 0.54) and accuracy in the JP partic-ipant group was higher for JP than for FR stimuli ($t(19)$ = 2.76, $p$ = .006, $d$ = 0.73). There were no other significant main effects, nor were there significant interactions between any of the other variables ($ps$ > .05).

We then analyzed each sentence condition separately with mixed 2 × 2 (Participant group × Language) ANOVAs to test whether LFEs on pitch shift detection could be observed across the degraded conditions (**Fig 4**). The LFE held in three of the four sentence types: In normal sentences, there was a significant main effect of Language ($F(1,42)$ = 6.25, $p$ = .016, $\eta_p^2$ = 0.13) and a significant interaction between Participant group and Language ($F(1,42)$ = 20.58,

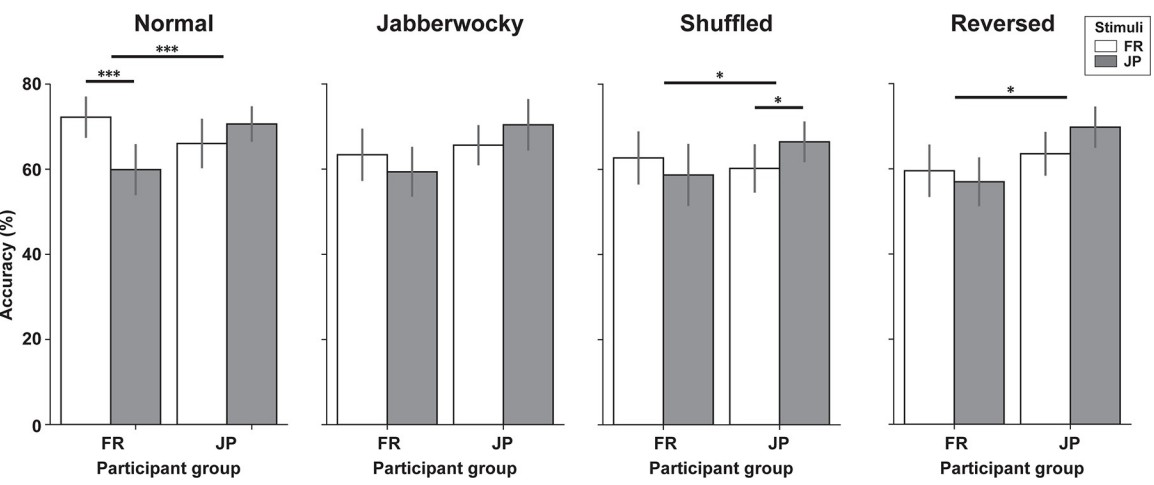

**Fig 4. Pitch shift detection: Accuracy in 2IFC detection of individually-calibrated pitch shifts, grouped by normal, jabberwocky, shuffled and reversed conditions (FR and JP sentences), for two groups of FR (N = 24) and JP (N = 20) speakers.** $^*p < .05$, $^{***}p < .001$, Bonferroni adjusted. Error bars, 95% CI.

$p < .001$, $\eta_p^2 = 0.33$). Paired $t$-tests showed that FR participants performed better in FR than JP sentences ($t(23) = 4.90$, $p < .001$, $\alpha_{Bonfer} = .025$, $d = 1.00$), but JP participants did not show a significant difference in performance between the FR and JP sentences ($t(19) = 1.66$, $p = .056$, $d = 0.45$).

For Jabberwocky sentences, there was a significant main effect of Participant group ($F(1,42) = 5.22$, $p = .027$, $\eta_p^2 = 0.11$). Paired $t$-tests did not show a significant difference between native and foreign language sentences in either participant group (FR participants: $t(23) = 1.25$, $p = .11$, $d = 0.29$; JP participants: $t(19) = 1.29$, $p = .11$, $d = 0.43$).

For shuffled sentences, there was a significant interaction between Participant group and language ($F(1,42) = 4.88$, $p = .033$, $\eta_p^2 = 0.10$). Paired $t$-tests showed that JP participants performed better in JP than FR sentences ($t(19) = 2.46$, $p = .012$, $d = 0.58$), but FR participants did not show a significant difference in performance between the FR and JP sentences ($t(23) = 1.09$, $p = .14$, $d = 0.26$).

For reversed sentences, there was a significant main effect of Participant group ($F(1,42) = 7.85$, $p = .008$, $\eta_p^2 = 0.16$) and a significant interaction between Participant group and Language ($F(1,42) = 4.47$, $p = .040$, $\eta_p^2 = 0.10$). Paired $t$-tests did not show a significant difference between native and foreign language sentences in either participant group (FR participants: $t(23) = 0.93$, $p = .18$, $d = 0.19$; JP participants: $t(19) = 2.01$, $p = .029$, $d = 0.61$).

## Discussion

Previous work on cross-cultural emotion recognition had left it difficult to determine whether native-language advantages arise from perceptual language-familiarity effects of the sort seen with speaker or face recognition or, more simply, from cultural differences in how these emotions are expressed. Here, to rule out such production differences, we used algorithmic voice transformations to create pairs of stimuli in the French and Japanese languages, which differed by exactly the same acoustical characteristics. Even though the cues were strictly identical in both languages, we found that they were better categorized emotionally (Experiment 1) and better detected (Experiment 2) when French and Japanese participants processed them in their native language.

Results from Experiment 1 provide evidence that production differences (or "speaker dialects" [12]) are not the sole drivers of in-group advantages in cross-cultural emotion perception. For example, even when we controlled happiness to be expressed with a precisely-calibrated pitch change of +50 cents in both languages, participants more accurately recognized such changes when they occurred in their native language. Critical to our manipulation is the fact that both groups reached above-chance emotion recognition on their normal native speech, showing that the computer-generated cues were salient and recognizable in both languages.

This native-language advantage can therefore only be explained by an interaction between the processes of encoding the linguistic or paralinguistic features of the familiar and non-familiar language and the processes responsible for extracting the emotional features important for the task—in short, by a perceptual learning effect of the kind already known for other-race face or speaker recognition [1–4]. Emotional cue extraction may be facilitated in the case of native-language (for example, better representation of what is phonological, and therefore better discrimination of what is incidental and expressive; see for example, [20]), or negatively affected in the case of a non-familiar language (for example, more effortful encoding, and therefore less resources available to process expressive cues; see for example, [34]), or both. Strikingly, the effect size of the native-language advantage in emotion recognition accuracy seen in Experiment 1 (FR: +8.0%, JP: +6.2% unbiased hit rate; **Fig 3**) is comparable with that of meta-studies of the in-group advantage in cross-cultural emotion recognition (+9.3% [18]), despite the fact that studies included in such meta-analyses typically subsumed both speaker- and listener-level influences on emotion recognition, and in particular differences in how the emotions were displayed by actors cross-culturally. This suggests that the perceptual LFE uncovered here is by no means a minor contributor in cross-cultural misperceptions, but rather explain a large proportion of such effects.

Although LFEs on emotion recognition were found for both participant groups, Experiment 1 also revealed several differences in how FR and JP participants processed foreign speech, as well as differences in recognition between emotion categories. First, while FR participants did not recognize emotional cues in normal JP above chance level, JP participants recognized emotional cues in normal FR less accurately than in normal JP, but still well above chance level. One possible cause of such asymmetry is the effect of linguistic closeness of English and French as members of the Indo-European language family and having influenced each other over history. While we confirmed that no JP participants had learned French before the experiment, Japanese students routinely learn English in the course of their education, and it is possible that familiarity with English facilitated emotion recognition in French (an effect also discussed by Scherer et al. about German and Dutch [12]). Second, irrespective of listener's language, we found that happiness was more distinct in its recognizability across languages (M = +12.3% unbiased hit rate) than afraid (M = +6.6%) and sadness (M = +2.3%). Although this result might be related to the modified transformation parameters in the current study from our previous study [22], it is consistent with previous meta-studies of in-group advantage in the voice modality (happiness: +17.5%; afraid: +12.9%; sadness: +7.6% [18]). Moreover, this result confirms the general notion that expressions of happiness are a lot less universal in the vocal modality [14, 17] than they are in the (overwhelmingly smile-driven) facial modality [18, 35], possibly because they rely on cues that require sufficiently accurate phonological representations of the target language to be extracted successfully. Interestingly, this would be the case, for example, of the smiling oro-facial gesture which, universal as it may be in the visual domain, translates acoustically to fine formant-related variations of vowel spectra which may require previous language exposure [36]. Finally, we found that the performance of JP participants did not differ between native- and foreign-language stimuli for the sad emotion,

although it did for happy and afraid. This may reflect either a lack of specificity of the sad voice transformation, a lack of specificity of the sad response category, or both. It is possible that, while the computer-generated cues used here were generally appropriate for all emotions in both languages, they were comparatively further away from the cultural norm of how sadness is expressed in the JP language. For instance, Lewis et al. (2010) have reported that Japanese children expressed less sadness but more embarrassment than American children after failing at games [37], possibly due to the cultural norm of emotion functions in Japan that focus on relationships with others. Yet, if these cues were wildly inappropriate for the JP culture, one would predict that JP participants would have similar difficulties recognizing such inappropriate cues when applied to FR stimuli, in which case we would see a main effect of language at a lower absolute level of recognition, but not a negation of the LFE for that emotion. Our data instead suggests that the processing of computer-generated cues was only impaired at a high-level of cultural expertise, indicating that any such ambiguity was masked when JP participants processed FR stimuli. This data is consistent with earlier reports of JP participants responding "sad" when presented happy JP stimuli more often than the other groups of participants [22], and may result from different boundaries between emotional terms: it is possible that the cues manipulated in the happy effect spanned a larger proportion of the vocal expressions referred to as "sad" (*kanashimi*) in Japanese than the proportion of expressions referred to as "sad" (*triste*) in French.

While previous literature has already suggested that perceptual-learning effects could drive in-group advantages in cross-cultural emotional voice perception [14, 38], it remained unclear at what level such linguistic or paralinguistic processes interfered with emotional cue recognition: non-native language stimuli differ both on lower-level auditory and phonological features, such as formant distributions and phoneme categories, mid-level prosodic representations, such as supra-segmental patterns for stress and prominence, but also higher-level syntax and semantics. In Experiment 1, we contrasted transformations applied to normal stimuli with identical transformations applied to three degraded conditions, breaking semantics (grammatical sentences with non-words, or Jabberwocky [27]), syntax (shuffled Jabberwocky), and supra-segmental patterns (reversed speech, see for example, [4]), and found evidence of LFEs (significant interaction of Language × Participant group) in all three conditions. This suggests that neither semantics, syntax, or supra-segmental patterns are necessary for this in-group advantage to emerge. This finding is consistent with LFEs in speaker recognition, which are preserved with reversed-speech [4] and absent in participants with impaired phonological representations [39], but is especially striking in the case of emotional prosody, which is in well-known interaction with syntax [40] and semantics [41]. That a native-language advantage to emotion recognition subsisted even in non-intelligible reversed speech suggests that the processes impacted by language-familiarity operate primarily at the segmental level, and that it is the listeners' lack of phonological familiarity with the individual speech sounds of the other language that impairs their processing of higher-level emotional cues.

The results of Experiment 2 partially replicate those of Experiment 1, and extend them to the lower-level task of detecting pitch shifts in running speech, a mechanism which subtends emotional prosodic processing [42]. As in Experiment 1, a significant interaction between Participant group and Language indicated a clear LFE, in which participants detected sudden pitch changes more accurately in the middle of sentences in their native language, in both participant groups. Although the LFE did not reach statistical significance across all of degraded sentence conditions (see below), this response pattern was consistent with Experiment 1, and suggests that the effect of language familiarity on emotion recognition is at least partly based on non-emotional and non-cultural differences in auditory capacities such as pitch processing.

Consistent with Experiment 1, this advantage persisted in two of the three types of stimulus degradation (shuffled and reversed), showing that neither semantics, syntax or supra-

segmental patterns are necessary for this pitch-processing advantage to emerge. However, pitch LFEs appeared somewhat less robust to stimulus degradation than emotion LFEs, as post-hoc tests only showed language differences in the normal sentence condition for FR participants and in the shuffled sentence condition for JP participants. Because pitch change detection is a lower-level, more robust auditory process than the extraction and categorization of emotional cues, it is possible that accuracy effects at that level are more difficult to observe experimentally than for emotions. These results mirror those of a prior study by Perrachione and colleagues [43] who found evidence for LFEs in talker identification in forward, but not in time-reversed speech. Similarly, in an effort to replicate the study by Fleming and colleagues [4], Perrachione et al. [44] tried but failed to find an LFE in perceptual dissimilarity judgments in time-reversed speech. Whether LFEs in talker identification and discrimination are driven by phonological or linguistic processes is still a topic of debate and it has been proposed that reliance on phonological or linguistic processes varies according to the specific experimental paradigm (for example, objective talker identification vs. subjective judgments on talker dissimilarity [44]). In this light it is possible that the degradation of the stimuli presented in the current studies affected the more gestalt-like emotional judgments in a different manner than lower-level pitch perception.

It is worth considering the experimental settings that may have influenced the current results. First, although the original English sentences were found to be emotionally neutral, we did not test the neutrality of the translated sentences in Japanese and French, and this setting could introduce a bias in the selection of certain emotion categories. However, we corrected such bias (if any) by using unbiased hit rates. This measure corrects the original hit rate using by taking into account how often participants select certain emotion categories regardless of the presented (actual) emotion categories. Therefore, we believe that such confusion (if any) would not affect the current results. Second, different transformation parameters from our previous study [22] might have affected emotion recognition performance in Experiment 1. For example, a smaller pitch shift value might reduce perceived sadness. However, there is a trade-off between transformation intensity and perceived naturalness [22]; we need to look for optimal parameters within which naturalness is not disrupted. Third, we did not manipulate perceived loudness and cannot make any argument about its impact on the results. Indeed, sound level and its variability may play a role in conveying vocal emotion [17]. To minimize such an effect and to focus on a few acoustic parameters, we normalized the root mean square intensity across all stimuli in the current study. However, it is possible that, by manipulating other acoustic parameters such as sound level, vocal emotion may become more salient and lead to higher recognition rates. Taken together, the present results nevertheless provide strong evidence that, for the most part, it is the listeners' lack of phonological familiarity with the individual speech sounds of the other language, and not with its syntax, semantics, or cultural production norms, which drive in-group advantages in cross-cultural emotion perception, and suggest that these effects are driven by differences in low-level auditory processing skills, such as the detection of sudden pitch shifts. Language exposure is known to improve the efficiency of timbre/formantic auditory representations, leading young infants to develop a native-language advantage to recognize phoneme categories [10] and speaker identities [11] by about 6–12 months. Similarly, in musicians, years of musical training correlate with more efficient subcortical encoding of auditory roughness and spectral complexity [45], and finer subcortical tracking of amplitude contours in a musician's own instrument [46]. In a related study, Krishnan and colleagues [47] recorded brainstem frequency-following responses (FFRs) elicited by four Mandarin tones in two groups of Mandarin Chinese and English speakers, and found that they tracked pitch contours more accurately in the Chinese group. It is therefore possible that similarly enhanced representations of the acoustical characteristics of native speech

sounds at the cortical or subcortical level facilitated the processing of pitch and emotional cues in our participants. Future work could extend this work by looking at neurophysiological indices of subcortical (auditory brainstem responses) or early cortical (mismatch negativity) auditory processing of pitch changes, and how they are modulated by language experience. Future work should also measure pitch cue processing and emotion recognition performance in the same individuals, to provide a stronger link between both types of processing [42].

More generally, we find it quite remarkable that for some individual participants, something as basic as the ability to detect a +30 cents increase of pitch in normal speech is influenced by their familiarity with the sounds of a given language. Such a low-level effect is bound to have important consequences down the auditory processing line on all judgments based on pitch variations: not only emotional expression (Experiment 1), but also syntactic or sentence mode information (for example, whether a sentence is interrogative or declarative), stress (for example, on what word is the sentence's focus) or attitudinal content (for example, whether a speaker is confident or doubtful) [48]. When pitch changes are already difficult to process in a foreign language, a lot more can be expected to be *lost in translation*. Future work could use reverse-correlation paradigms to investigate whether language-familiarity effects translate to qualitatively different mental representations of pitch contours or different levels of internal noise linked to these representations [49].

## Supporting information

**S1 File. Example audio files used in Experiment 1.**
(ZIP)

## Acknowledgments

The authors thank Maël Garnotel, Gabriël Vogel, and Michiko Asano for their help with data collection.

## Author Contributions

**Conceptualization:** Tomoya Nakai, Laura Rachman, Jean-Julien Aucouturier.

**Data curation:** Tomoya Nakai, Pablo Arias Sarah.

**Formal analysis:** Tomoya Nakai.

**Funding acquisition:** Kazuo Okanoya, Jean-Julien Aucouturier.

**Methodology:** Tomoya Nakai, Laura Rachman, Pablo Arias Sarah.

**Resources:** Jean-Julien Aucouturier.

**Software:** Laura Rachman, Jean-Julien Aucouturier.

**Supervision:** Kazuo Okanoya.

**Validation:** Tomoya Nakai, Laura Rachman, Jean-Julien Aucouturier.

**Writing – original draft:** Tomoya Nakai, Laura Rachman, Jean-Julien Aucouturier.

**Writing – review & editing:** Tomoya Nakai, Kazuo Okanoya, Jean-Julien Aucouturier.

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
