## [Decision Letter · Decision Letter 0]

12 Jan 2023

PONE-D-22-22873Algorithmic voice transformations reveal the phonological basis of language-familiarity effects in cross-cultural emotion judgmentsPLOS ONE

Dear Dr. Nakai,

Thank you for submitting your manuscript to PLOS ONE. After careful consideration, we feel that it has merit but does not fully meet PLOS ONE’s publication criteria as it currently stands. Therefore, we invite you to submit a revised version of the manuscript that addresses the points raised during the review process.

Thank you for submitting your manuscript to PLOS ONE. After careful consideration, we feel that it has merit but does not fully meet PLOS ONE's publication criteria as it currently stands. Therefore, we invite you to submit a revised version of the manuscript that addresses the points raised during the review process. While the manuscript was overall found satisfactory, there are a series of requests for clarifications that need to be carefully addressed to clear out the remaining concerns of the reviewers. I think addressing these points will strengthen the manuscript. Reviewer 1 raised two points I think need addressing before publication. These are 1) how the parameters of each transformation were identified (see reviewer's 1 comment regarding "Auditory manipulation"), and 2) how the transformations applied to the recordings may have impacted the results (see the comment regarding "Stimuli transformations"). Reviewer 2 requested further clarification on 1) the sad response category for Japanese participants and the cultural norm of how sadness is expressed in Japanese, and 2) details on the task method, particularly the clicking error rate and data possibly excluded because of error in clicks. 

We look forward to receiving your revised manuscript.

Kind regards,

Federica Cavicchio

Academic Editor

PLOS ONE

and https://journals.plos.org/plosone/s/file?id=ba62/PLOSOne_formatting_sample_title_authors_affiliations.pdf.

“All the data in this experiment was collected at the Centre Multidisciplinaire des Sciences Comportementales Sorbonne Université–Institut Européen d’Administration des Affaires (INSEAD), the University of Tokyo, and the Center for Information and Neural Networks (Japan). The authors thank Maël Garnotel, Gabriël Vogel, and Michiko Asano for their help with data collection. This work was supported by ERC Grant StG 335536 CREAM, Fondation pour l'Audition FPA RD-2018-2, ANR REFLETS and SEPIA, and JSPS KAKENHI Grant Number 20H05023 and 20K07718.”

“This work was supported by ERC Grant StG 335536 CREAM, Fondation pour l'Audition FPA RD-2018-2, ANR REFLETS and SEPIA, and JSPS KAKENHI Grant Number 20H05023 and 20K07718. The funders had no role in study design, data collection and analysis, decision to publish, or preparation of the manuscript.”

Additional Editor Comments:

Thank you for submitting your manuscript to PLOS ONE. After careful consideration, we feel that it has merit but does not fully meet PLOS ONE's publication criteria as it currently stands. Therefore, we invite you to submit a revised version of the manuscript that addresses the points raised during the review process. While the manuscript was overall found satisfactory, there are a series of requests for clarifications that need to be carefully addressed to clear out the remaining concerns of the reviewers. I think addressing these points will strengthen the MS. Reviewer 1 raised two points I think need addressing before publication. These are 1) how the parameters of each transformation were identified (see reviewer's 1 comment regarding "Auditory manipulation"), and 2) how the transformations applied to the recordings may have impacted the results (see the comment regarding "Stimuli transformations"). Reviewer 2 requested further clarification on 1) the sad response category for Japanese participants and the cultural norm of how sadness is expressed in Japanese, and 2) details on the task method, particularly the clicking error rate and data possibly excluded because of error in clicks.

Reviewers' comments:

Reviewer's Responses to Questions

**Comments to the Author**

1. Is the manuscript technically sound, and do the data support the conclusions?

Reviewer #1: Yes

Reviewer #2: Yes

2. Has the statistical analysis been performed appropriately and rigorously? 

Reviewer #1: Yes

Reviewer #2: Yes

3. Have the authors made all data underlying the findings in their manuscript fully available?

Reviewer #1: Yes

Reviewer #2: Yes

4. Is the manuscript presented in an intelligible fashion and written in standard English?

Reviewer #1: Yes

Reviewer #2: Yes

5. Review Comments to the Author

Reviewer #1: Thank you for the opportunity to review your paper. This paper presents new research into cross cultural emotion recognition in voice using software to ensure consistency between stimuli. Consistency in vocal stimuli presents a significant challenge for emotion research involving the voice, and this paper is novel both in its findings and in demonstrating how the DAVID software can be applied allowing the experimenter more control over their stimuli.

Overall, I think this paper will make an excellent contribution to the journal. However, there are several points that would benefit from revision/clarification before publication. I itemize these below using line numbers to hopefully make these points more easily identifiable in the manuscript.

GENERAL POINTS REGARDING FORMATTING, EXPRESSION, CLARITY ETC.

Line 42 – 43, additional brackets not required around citation “…(other-race effect; (Shapiro and Penrod 1986; Meissner and Brigham 2001)),…”. Same for citation in like 44. There are other places in the paper where additional brackets appear in in text citations. Recommend checking for these (seems to just be an artifact of the reference management software?).

Lines 48 – 52, “These familiarity effects are thought to result primarily from perceptual learning: differential exposure warps an observer's perceptual space to better accommodate distinctions that are informative to discriminate common in-group items, with the result that comparatively less-common out-group items are encoded less efficiently …”. Recommend rephrasing this (perhaps splitting into two sentences?) to make the meaning clearer.

Line 59 – 63, “…high or low, a given phoneme bright or dark, whether a specific pitch inflection is expressive or phonological…” recommend moving the citation to the end of the sentence.

Line 61, “…all would appear…” recommend rephrasing to “…would all appear…”.

Line 64-66, issues with the formatting of the in text citation, are Scherer etc all cited in Elfenbine and Ambady 2002, and Lukka and Elfenbine 2021? Recommend reformatting the citation to make this clear.

Line 71, recommend changing “…but such evidence is mixed: Scherer and colleagues…” to “…but such evidence is mixed. Scherer and colleagues” so that Scherer begins a new sentence.

Line 102, here the emotions examined are referred to as “happiness, sadness, and fear/anxiety” but in the rest of the paper “afraid” was sometimes used to describe the emotion of fear/anxiety. This terminology should be aligned. What terms were presented to the participants in the experiment? Recommend using these. For example, if afraid was the term used in the experiment (or its translation) then used afraid in the paper rather than fear/anxiety?

Line 45, 73 and other places, recommend changing “e.g.” to “for example” throughout the paper.

Line 87, would recommend moving “DAVID” out of the brackets to make it clear that this is the name of the software. For example, “…processing software called DAVID (Rachman et al. 2018) to ..”

Line 132, remove bracket after “Table 1)…”?

Line 203, “…and JP (N = 21, 1 excluded)…” is that 21 with the 1 excluded or 20, plus 1 excluded?

Line 263, recommend rephrasing “…provide unequivocal evidence that…” to “…provide evidence that…”

Line 297, missing bracket after citation.

Line 300 – 301, check citation formatting with page numbers?

Line 334, sometimes “suprasegmental” is used, and other times “supra-segmental” is used. Recommend standardizing throughout

Line 381, recommend changing “6-12mo” to “6-12 months”

Line 395, “More generally, we find it quite remarkable that something as basic as the ability to detect a +30 cents increase of pitch in normal running speech is influenced by one observer's familiarity with the sounds of a given language” Does this refer to a single participant’s ability to detect +30 cents? Or an average threshold across participants?

Line 401 – 402, perhaps change “When already pitch changes are difficult …” to “When pitch changes are already difficult…”?

Line 416, were the sentences taken from Russ Gur, and Bilker 2008? If so, consider stating this explicitly.

Line 437, “In addition, we used four male and four female normal Swedish (SE) recordings from a previous study (Rachman et al. 2018).” Were these recordings of the same sentences recorded by the JP and FR speakers in this study?

Line 443, not sure that the URL needs brackets here?

Line 475, recommend replacing “complained” with “reported”

Line 479, delete repeated word “recordings”

Would recommend considering citing your dataset in the references too, following citation format such as that found at: https://www.nlm.nih.gov/bsd/uniform_requirements.html#item43. This may make it easier for your dataset citation counts to be captured.

Lines 494 – 495, “Visual stimuli were displayed on a laptop-computer screen,…” was this the written out sentences? Recommend explaining what these visual stimuli were and, if the visual stimuli was the written out sentences, why they were used.

SPECIFIC COMMENTS REGARDING METHODS, FINDINGS, INTERPRETATION

Sentences: Was the neutrality of the sentences validated for JP and FR translations? For example, a pre test to check that the JP translation of “I forgot my jacket” was also considered neutral? If not, recommend stating this explicitly and discussing why a pre test was not conducted/if there is any reason to think the translations might be less neutral in JP/FR than in English.

Auditory manipulation:

Line 441 – 462, including Table 2: Would recommend moving the sentence beginning on line 456 “These values were based on previous studies…” to the end of the first paragraph (and rephrasing appropriately) to make it clear up front that the transformations have been validated already.

If I understood correctly, I understand that the transformations described in were based on Aucouturier et al. 2016 and Rachman et al. 2018. I think it would be useful to state which transformations were drawn from which studies. For example, if I am understanding correctly, when I look at the transformations tested in Rachman et al. 2018 I see that the transformations used in the present paper don’t exactly match the low/medium/high levels tested in Rachman et al. 2018, Table 2. In this paper, pitch is shifted +50 cents for happy, which matches Rachman pitch shift for “high” happy, but in this paper sad is shifted -50 cents which does not exactly match any of the transformations from Rachman Table 2. Also, is there a reason that vibrato depth was not adjusted for male and female speakers in this study but it was in Rachman et al? It would be useful to explain why any deviations from the validated transformations were made. This could be addressed in the paragraph beginning line 446. Recommend also stating here that the categories of happy, sad, afraid were used due to the validation in Rachman et al (sorry if this is already in the paper and I missed it).

Discussion: Lines 297 – 305, beginning “Second, irrespective of listener’s language,…” to what degree might this be related to the intensity of the transformations? Is it possible that the afraid and sad stimuli were not as intensely afraid and sad as happy stimuli? This relates to the point directly above and how the value of the transformations were selected.

Stimuli transformations: It’s always possible that results might be impacted by the way stimuli were created. Given this study used computer software to transform voice for the stimuli, DAVID, it would be worth including a paragraph in the discussion exploring how the use of DAVID might have impacted results. For example, are there differences in perceived naturalness across the emotions which are an artifact of the transformations that may have interfered with the results? Do these results say anything about the role of loudness in emotion perception given loudness is not part of the DAVID transformation configurations? This would strengthen the paper and help the reader identify areas in which ecological validity could be improved.

Instructions: Can you include in an appendix or in the OSF repository the actual study protocol and instructions? That is, the actual instructions given to participants in the respective languages? This might be useful in understanding the implications of the disclaimer beginning line 496-497 that “Response categories in the recognition task for the French group…”. It may help the reader understand the possible implications of this difference between groups. As noted in the present paper, the different translations of the emotion words might have impacted results (sorry if this is already included in the OSF repo and I have missed it).

Reviewer #2: The paper investigates the effect of familiarity with a language phonology in emotion judgements, using audio processing software (DAVID) for testing speakers of different languages on identical acoustic cues. While most previous studies tested language familiarity effects using stimuli created by human actors, the present study uses stimuli that have the exact same modifications, and yet sound natural, to reduce biases which may be introduced by the variability of the human voice in the speech signal. The paper is clear, written in good English and well organized. The literature review is exhaustive. The study is scientifically sound, the experiments have been carried out in a rigorous manner, with an appropriate methodology, and an appropriate number of participants and stimuli. The data and the experimental results support the conclusions.

I have a few comments/requests for clarification:

@310-313: the authors talk about the lack of specificity of the sad response category and how the JP participants were comparatively further away from the cultural norm of how sadness is expressed in the JP language. This is very interesting, but somewhat obscure for people that are not familiar with this subject. Providing a more detailed explanation of this issue would benefit the reader and help them get a better understanding of lines 320-321.

@489-491: "After hearing the two utterances, participants (all right-handed) were instructed to answer whether the second utterance sounded happy, sad, afraid, or neutral by pressing one of four keys (“S”, “D”, “F”, and “G”) with their left fourth, third, second, and first finger, respectively". This task may have been rather confusing and/or somewhat difficult for the participants: they had to memorize the association of the emotions happy, sad, afraid, or neutral, with 4 keyboard keys that had no meaning association with them, and had to click on the keys with 4 different fingers. Did the participants ever get confused? Did this impact the experiment in any way? Had some result to be excluded because of this? Please provide further details.

@497-499: why was there a difference in the presentation of terms in the 2 language groups (i.e., the Fr were given English terms and the JP were given JP terms)? The reason for this difference in the procedure needs to be explained. Also, details need to be provided to let the reader know whether this difference affected the results in any way.

Minor corrections in the text:

@line 280: 'in' is missing in: accuracy seen Experiment 1

@line 402: in 'lost in translation' the initial 'l' is not in italic

@line 479: The word recordings is repeated twice in: All sentence recordings recordings described...

6. PLOS authors have the option to publish the peer review history of their article (what does this mean?). If published, this will include your full peer review and any attached files.

Reviewer #1: No

Reviewer #2: No

---

## [Author Response · Author response to Decision Letter 0]

20 Feb 2023

We greatly appreciate the reviewers’ time and effort in reviewing our manuscript and providing helpful comments. Based on their comments, we have revised the original manuscript. Please find our responses to the comments below, along with the related changes in the revised manuscript.

Reviewer #1: 

Comment 1-1: GENERAL POINTS REGARDING FORMATTING, EXPRESSION, CLARITY ETC.

Line 42 – 43, additional brackets not required around citation “…(other-race effect; (Shapiro and Penrod 1986; Meissner and Brigham 2001)),…”. Same for citation in like 44. There are other places in the paper where additional brackets appear in in text citations. Recommend checking for these (seems to just be an artifact of the reference management software?).

Response 1-1: We thank the reviewer for pointing out this citation error. This was caused by the citation management software. In the new manuscript, we have implemented PLOS ONE's citation style. Consequently, the text is now presented as follows:

“(other-race effect [1,2])”

Comment 1-2: Lines 48 – 52, “These familiarity effects are thought to result primarily from perceptual learning: differential exposure warps an observer's perceptual space to better accommodate distinctions that are informative to discriminate common in-group items, with the result that comparatively less-common out-group items are encoded less efficiently …”. Recommend rephrasing this (perhaps splitting into two sentences?) to make the meaning clearer.

Response 1-2: We apologize for the hard-to-read sentences. We have rephrased the text as follows:

“These familiarity effects are thought to result primarily from perceptual learning; differential exposure warps an observer's perceptual space to facilitate discrimination of common in-group items. Comparatively rare out-group items are thus encoded less efficiently [8,9].”

Comment 1-3: Line 59 – 63, “…high or low, a given phoneme bright or dark, whether a specific pitch inflection is expressive or phonological…” recommend moving the citation to the end of the sentence.

Response 1-3: We thank the reviewer for the suggestion. We have moved the citation to the end of the sentence as follows: 

“by acquired auditory representations optimized for the sounds of one language or another [16,17].”

Comment 1-4: Line 61, “…all would appear…” recommend rephrasing to “…would all appear…”.

Response 1-4: We thank the reviewer for the suggestion. We have corrected the text as follows:

“a given phoneme bright or dark, whether a specific pitch inflection is expressive or phonological would all appear to be advantaged”

Comment 1-5: Line 64-66, issues with the formatting of the in text citation, are Scherer etc all cited in Elfenbine and Ambady 2002, and Lukka and Elfenbine 2021? Recommend reformatting the citation to make this clear.

Response 1-5: We thank the reviewer for pointing out the format issue. Thompson and Balkwill (2006) and Pell et al. (2009) were not cited in Elfenbein and Ambady (2002) but in Laukka and Elfenbein (2021). To avoid misunderstandings, we only list the review articles in this sentence in the new manuscript. Also, as in response to Comment 1-1, we have implemented PLOS ONE's citation style. The text is written as follows in the new manuscript:

“Most cross-cultural data indeed reveal an in-group advantage for identifying emotions displayed by members of the same rather than a different culture (see [18,19] for a review).”

Comment 1-6: Line 71, recommend changing “…but such evidence is mixed: Scherer and colleagues…” to “…but such evidence is mixed. Scherer and colleagues” so that Scherer begins a new sentence.

Response 1-6: We thank the reviewer for the suggestion. We have changed the text as follows: 

“but such evidence is mixed. Scherer and colleagues found Dutch listeners better at decoding German utterances than listeners of other, less similar European and Asian languages [12].”

Comment 1-7: Line 102, here the emotions examined are referred to as “happiness, sadness, and fear/anxiety” but in the rest of the paper “afraid” was sometimes used to describe the emotion of fear/anxiety. This terminology should be aligned. What terms were presented to the participants in the experiment? Recommend using these. For example, if afraid was the term used in the experiment (or its translation) then used afraid in the paper rather than fear/anxiety?

Response 1-7: We apologize for the lack of consistency. In the experiment, we always used the term “afraid”. Therefore, we have changed the relevant texts as follows:

In Introduction,

“In a series of two experiments, we manipulate here both Japanese (JP) and French (FR) voices with the same set of parametric transformations by DAVID, so as to display emotions of happiness, sadness, and afraid (Experiment 1)”

In Discussion,

“Second, irrespective of listener’s language, we found that happiness was more distinct in its recognizability across languages (M = +12.3% unbiased hit rate) than afraid (M = +6.6%) and sadness (M = +2.3%).”

Comment 1-8: Line 45, 73 and other places, recommend changing “e.g.” to “for example” throughout the paper.

Response 1-8: We thank the reviewer for the suggestion. We have changed “e.g.” to “for example” throughout the manuscript. 

As for the line 73, we changed the text as follows:

“However, other studies found no differences in, for example, how Spanish listeners identified vocal emotions from the related English and German languages, and the unrelated Arabic [14],”

For the line 45, we think that we do not need “for example” here and thus removed it from this sentence.

Comment 1-9: Line 87, would recommend moving “DAVID” out of the brackets to make it clear that this is the name of the software. For example, “…processing software called DAVID (Rachman et al. 2018) to ..”

Response 1-9: We thank the reviewer for the comment. We have modified the text as follows:

“we used audio processing software called DAVID [22] to apply a fixed set of programmable acoustical/emotional transformations”

Comment 1-10: Line 132, remove bracket after “Table 1)…”?

Response 1-10: We thank the reviewer for pointing out our mistake. We have corrected the text as follows (note that the table number has been changed because we moved the Results section after the Materials and Methods section):

“(see Table 3, Bonferroni corrected across twelve comparisons).”

Comment 1-11: Line 203, “…and JP (N = 21, 1 excluded)…” is that 21 with the 1 excluded or 20, plus 1 excluded?

Response 1-11: We apologize for the lack of clarity. The number of JP participants used in the final analysis was 20. One participant was excluded from the original group of 21 participants. To avoid misunderstandings, we changed the text in Experiment 2: Pitch shift detection as follows:

“we therefore tested two new groups of FR (N = 24) and JP (N = 20)”

The exclusion of one JP participant has been described in the Exp. 2 - Pitch change detection > Participants subsection:

“One JP male participant was excluded from the analyses because he reported to have changed strategies between the calibration and the test phase”

Comment 1-12: Line 263, recommend rephrasing “…provide unequivocal evidence that…” to “…provide evidence that…”

Response 1-12: We thank the reviewer for this suggestion. We have modified the text as follows:

“Results from Experiment 1 provide evidence that production differences (or “speaker dialects” [12]) are not the sole drivers of in-group advantages in cross-cultural emotion perception.”

Comment 1-13: Line 297, missing bracket after citation.

Response 1-13: We thank the reviewer pointing out our mistake. We have corrected the text as follows:

“it is possible that familiarity with English facilitated emotion recognition in French (an effect also discussed by Scherer et al. about German and Dutch [12]).”

Comment 1-14: Line 300 – 301, check citation formatting with page numbers?

Response 1-14: We thank the reviewer for pointing out this formatting issue. We could not find a general guideline for the case where a single page of a particular article is referenced within the main manuscript. To avoid confusion, we have modified the relevant text in the Discussion as follows:

“it is consistent with previous meta-studies of in-group advantage in the voice modality (happiness: +17.5%; afraid: +12.9%; sadness: +7.6% [18])”

Similarly,

“comparable with that of meta-studies of the in-group advantage in cross-cultural emotion recognition (+9.3% [18])”

Comment 1-15: Line 334, sometimes “suprasegmental” is used, and other times “supra-segmental” is used. Recommend standardizing throughout

Response 1-15: We thank the reviewer for pointing out the inconsistent wording. We use “supra-segmental” throughout the revised manuscript.

In particular, we have changed the following text in Abstract:

“which disturbed semantics, syntax, and supra-segmental patterns, respectively.”

In the Discussion section, we changed as follows:

“(grammatical sentences with non-words, or Jabberwocky [31]), syntax (shuffled Jabberwocky), and supra-segmental patterns”

Comment 1-16: Line 381, recommend changing “6-12mo” to “6-12 months”

Response 1-16: We thank the reviewer for the comment. We have corrected the text as follows:

“leading young infants to develop a native-language advantage to recognize phoneme categories [10] and speaker identities [11] by about 6-12 months.”

Comment 1-17: Line 395, “More generally, we find it quite remarkable that something as basic as the ability to detect a +30 cents increase of pitch in normal running speech is influenced by one observer's familiarity with the sounds of a given language” Does this refer to a single participant’s ability to detect +30 cents? Or an average threshold across participants?

Response 1-17: This refers to some individual participants who were able to detect a pitch increase of 30 cents. To clarify this point, we have added the following text in the Discussion section:

“More generally, we find it quite remarkable that for some individual participants, something as basic as the ability to detect a +30 cents increase of pitch in normal speech is influenced by their familiarity with the sounds of a given language.”

Comment 1-18: Line 401 – 402, perhaps change “When already pitch changes are difficult …” to “When pitch changes are already difficult…”?

Response 1-18: We thank the reviewer for the comment. We have corrected the text as follows:

“When pitch changes are already difficult to process in a foreign language”

Comment 1-19: Line 416, were the sentences taken from Russ Gur, and Bilker 2008? If so, consider stating this explicitly.

Response 1-19: Yes, the original sentences were taken from Russ et al. (2008). In the new manuscript, we explicitly state it in Recordings and Experimental Apparatus subsection as follows:

“translated from the same four semantically-neutral English sentences taken from Russ et al. (2008) [43].”

Comment 1-20: Line 437, “In addition, we used four male and four female normal Swedish (SE) recordings from a previous study (Rachman et al. 2018).” Were these recordings of the same sentences recorded by the JP and FR speakers in this study?

Response 1-20: We thank the reviewer for commenting on this important point. These were not exactly the same sentences as in the JP and FR sentences, but were also taken from Russ et al. (2008). We did not choose the same Swedish sentences because some of them were long and could not be cut into the 1.5 s stimuli. However, we believe that this setting does not affect the results because none of the participants could understand the meaning of the Swedish sentences, and the Swedish sentences are not the main target of cross-cultural analyses.

To explain the above point, we have added the following text in Recordings and Experimental Apparatus subsection:

“Although these SE sentences did not contain the same semantic information as JP and FR sentences, we considered them as a baseline control for unfamiliar language for both participant groups.”

Comment 1-21: Line 443, not sure that the URL needs brackets here?

Response 1-21: We thank the reviewer for the suggestion. We have removed the bracket from the following text in Audio Manipulation Algorithm subsection:

“The audio recordings were manipulated with the software platform DAVID [22], available at https://forum.ircam.fr/projects/detail/david/.”

Comment 1-22: Line 475, recommend replacing “complained” with “reported”

Response 1-22: We thank the reviewer for this suggestion. We have changed the text as follows:

“one Japanese participant was excluded because they reported that they could hear the auditory stimuli only from one side of the headphone.”

Comment 1-23: Line 479, delete repeated word “recordings”

Response 1-23: We thank the reviewer for pointing out our mistake. We have corrected the text as follows:

“All sentence recordings described above (incl. reverse JP, reverse FR, and normal SE) were processed with DAVID”

Comment 1-24: Would recommend considering citing your dataset in the references too, following citation format such as that found at: https://www.nlm.nih.gov/bsd/uniform_requirements.html#item43. This may make it easier for your dataset citation counts to be captured.

Response 1-24: We thank the reviewer for this suggestion. We have made our data and code publicly available and added in the references [49]. We have therefore changed the following text in the Data availability subsection:

“All analysis codes and data are available from Open Science Framework (https://osf.io/4py25/) [49].”

We have added the following citation:

“Nakai T, Rachman L, Arias P, Okanoya K, Aucouturier JJ. Data & Code: Algorithmic voice transformations reveal the phonological basis of language-familiarity effects in cross-cultural emotion judgments. Available from: https://osf.io/4py25/

Comment 1-25: Lines 494 – 495, “Visual stimuli were displayed on a laptop-computer screen,…” was this the written out sentences? Recommend explaining what these visual stimuli were and, if the visual stimuli was the written out sentences, why they were used.

Response 1-25: We presented only four different emotion labels with corresponding keys and instructions. Auditory stimuli were not presented visually. To avoid confusion, we added the following text to the Procedure subsection:

“After hearing the two utterances, four emotion labels with corresponding keys were presented visually (“[S] happy”, “[D] sad”, “[F] afraid”, “[G] identical”)”

See also the response to Comment 1-30 regarding uploading instruction texts to OSF.

Comment 1-26: SPECIFIC COMMENTS REGARDING METHODS, FINDINGS, INTERPRETATION

Sentences: Was the neutrality of the sentences validated for JP and FR translations? For example, a pre test to check that the JP translation of “I forgot my jacket” was also considered neutral? If not, recommend stating this explicitly and discussing why a pre test was not conducted/if there is any reason to think the translations might be less neutral in JP/FR than in English.

Response 1-26: We appreciate the reviewer's comment on this important point. We did not test the neutrality of translated sentences. Russ et al. (2008) created neutral English sentences based on the words they contained, and the sentences used in our study are simple and consist almost entirely of directly translated words. Therefore, we assume that the FR and JP sentences in the current study did not convey any additional emotional content.

Even if there were an intrinsic bias in the neutrality of the sentences, it would not affect the current results because we examined the language familiarity effect using the unbiased hit rate. The unbiased hit rate can correct the tendency (if any) to select a particular emotion category for the neutral sentences for each language. Therefore, even if participants found the FR or JP sentences less neutral, such a bias would not show up in our results.

To explain the above points, we have added the following text in Discussion section:

“It is worth considering the experimental settings that may have influenced the current results. First, although the original English sentences were found to be emotionally neutral, we did not test the neutrality of the translated sentences in Japanese and French, and this setting could introduce a bias in the selection of certain emotion categories. However, we corrected such bias (if any) by using unbiased hit rates. This measure corrects the original hit rate using by taking into account how often participants select certain emotion categories regardless of the presented (actual) emotion categories. Therefore, we believe that such confusion (if any) would not affect the current results.”

Comment 1-27: Auditory manipulation:

Line 441 – 462, including Table 2: Would recommend moving the sentence beginning on line 456 “These values were based on previous studies…” to the end of the first paragraph (and rephrasing appropriately) to make it clear up front that the transformations have been validated already.

If I understood correctly, I understand that the transformations described in were based on Aucouturier et al. 2016 and Rachman et al. 2018. I think it would be useful to state which transformations were drawn from which studies. For example, if I am understanding correctly, when I look at the transformations tested in Rachman et al. 2018 I see that the transformations used in the present paper don’t exactly match the low/medium/high levels tested in Rachman et al. 2018, Table 2. In this paper, pitch is shifted +50 cents for happy, which matches Rachman pitch shift for “high” happy, but in this paper sad is shifted -50 cents which does not exactly match any of the transformations from Rachman Table 2. Also, is there a reason that vibrato depth was not adjusted for male and female speakers in this study but it was in Rachman et al? It would be useful to explain why any deviations from the validated transformations were made. This could be addressed in the paragraph beginning line 446. Recommend also stating here that the categories of happy, sad, afraid were used due to the validation in Rachman et al (sorry if this is already in the paper and I missed it).

Response 1-27: We appreciate the reviewer for pointing out this issue. We acknowledge that the transformation parameters in the current study do not match those in Rachman et al. (2018). The pitch shift of −50 cents in the sad voices was chosen to mirror that in the happy voices. We applied the same vibrato parameters to male and female voices because we considered that the original vibrato parameters were not effective enough for some of the female JP voices, and that too much modulation of the male voices would decrease naturalness. These parameter changes, however, do not impact our conclusion, because the most important point in the current study was that we used the same transformation parameters for both JP and FR stimuli. 

Based on the reviewer’s suggestion, we have moved the regarding sentence to the end of the first paragraph in the Audio Manipulation Algorithm subsection and added explanations about parameter changes as follows:

“For the current experiment, we used predetermined pitch shift, vibrato, inflection, and filtering transformations designed to evoke happy, sad, and afraid expressions. Table 2 describes the transformation parameter values used in this study, which were applied identically to both Japanese and French-language stimuli. These values were based on previous studies [22,23] that validated that they sounded both natural and recognizable in both JP and FR. Note that we modified some of the parameters from Rachman et al. [22]; the pitch shift in the sad voices was set to −50 cents to mirror that in the happy voices. We also applied the same vibrato parameters to male and female voices because the original vibrato parameters would not have enough effect on some of the female JP voices. Examples of the manipulations are illustrated in Figure 1, and in supplementary audio files.”

Comment 1-28: Discussion: Lines 297 – 305, beginning “Second, irrespective of listener’s language,…” to what degree might this be related to the intensity of the transformations? Is it possible that the afraid and sad stimuli were not as intensely afraid and sad as happy stimuli? This relates to the point directly above and how the value of the transformations were selected.

Response 1-28: We agree with the reviewer that the different transformation parameters in the current study may have resulted in a smaller effect in afraid and sad conditions. We have discussed this possibility as follows:

“Although this result might be related to the modified transformation parameters in the current study from our previous study [22], it is consistent with previous meta-studies of in-group advantage in the voice modality (happiness: +17.5%; afraid: +12.9%; sadness: +7.6% [18]). Moreover, this result confirms the general notion that expressions of happiness are a lot less universal in the vocal modality [14,17]”

We have also added the following limitation in the Discussion section: 

“Second, different transformation parameters from our previous study [22] might have affected emotion recognition performance in Experiment 1. For example, a smaller pitch shift value might reduce perceived sadness. However, there is a trade-off between transformation intensity and perceived naturalness [22]; we need to look for optimal parameters within which naturalness is not disrupted.”

Comment 1-29: Stimuli transformations: It’s always possible that results might be impacted by the way stimuli were created. Given this study used computer software to transform voice for the stimuli, DAVID, it would be worth including a paragraph in the discussion exploring how the use of DAVID might have impacted results. For example, are there differences in perceived naturalness across the emotions which are an artifact of the transformations that may have interfered with the results? Do these results say anything about the role of loudness in emotion perception given loudness is not part of the DAVID transformation configurations? This would strengthen the paper and help the reader identify areas in which ecological validity could be improved.

Response 1-29: We thank the reviewer for this suggestion. In our previous study (Rachman et al., 2018), we assessed the perceived naturalness of transformed voices, and it was comparable to authentic speech (but we also found that there is a trade-off between transformation intensity and perceived naturalness). This was discussed in the Introduction:

“DAVID was validated in multiple languages, finding that transformed emotions were recognized at above-chance levels when applied to either French, English, Swedish, or Japanese utterances, and with a naturalness comparable to authentic speech [22].”

As for the perceived loudness, the current results do not say anything about its effect on emotion recognition because we did not manipulate and equalize it across the stimuli. Indeed, previous studies have reported that sound level and its variability play a role in conveying vocal emotion, e.g. afraid can be characterized by overall low sound level and high sound level variability, happy by medium-high sound level, and sadness by low sound level and low variability (Juslin & Laukka 2003, Psychological Bulletin). To minimize such an effect and to focus on a few acoustic parameters, we normalized the root mean square intensity across all stimuli. However, it is possible that, by manipulating other acoustic parameters such as sound level, vocal emotion may become more salient and lead to higher recognition rates.

To explain this point, we have added the following text in Discussion section:

“Third, we did not manipulate perceived loudness and cannot make any argument about its impact on the results. Indeed, sound level and its variability may play a role in conveying vocal emotion [17]. To minimize such an effect and to focus on a few acoustic parameters, we normalized the root mean square intensity across all stimuli in the current study. However, it is possible that, by manipulating other acoustic parameters such as sound level, vocal emotion may become more salient and lead to higher recognition rates.”

Comment 1-30: Instructions: Can you include in an appendix or in the OSF repository the actual study protocol and instructions? That is, the actual instructions given to participants in the respective languages? This might be useful in understanding the implications of the disclaimer beginning line 496-497 that “Response categories in the recognition task for the French group…”. It may help the reader understand the possible implications of this difference between groups. As noted in the present paper, the different translations of the emotion words might have impacted results (sorry if this is already included in the OSF repo and I have missed it).

Response 1-30: Based on the reviewer’s suggestion, we have added instruction documents to the OSF repository (“Instruction_JP.txt,” “Instruction_FR.txt”).

Reviewer #2: 

Comment 2-1: @310-313: the authors talk about the lack of specificity of the sad response category and how the JP participants were comparatively further away from the cultural norm of how sadness is expressed in the JP language. This is very interesting, but somewhat obscure for people that are not familiar with this subject. Providing a more detailed explanation of this issue would benefit the reader and help them get a better understanding of lines 320-321.

Response 2-1: We appreciate the reviewer’s interest in this point. Lewis et al. (Int J Behav Dev, 2010) have reported that Japanese children expressed less sadness but more embarrassment after failing at games. According to their interpretation, this result may be due to the cultural norm of emotion functions in Japan, which focus on relationships with others, in contrast to Western societies, which focus on monitoring and expressing personal feelings. In other word, “Japanese children’s socialization emphasizes a “we-self” in contrast to the “I-self” of Western societies” (Lewis et al. 2010).

To explain the above point, we have added the following text in Discussion section:

“For instance, Lewis et al. (2010) have reported that Japanese children expressed less sadness but more embarrassment than American children after failing at games [29], possibly due to the cultural norm of emotion functions in Japan that focus on relationships with others.”

We added the following citation:

“Lewis M, Takai-Kawakami K, Kawakami K, Sullivan MW. Cultural Differences in Emotional Responses to Success and Failure. Int J Behav Dev. 2010;34: 53–61.”

Comment 2-2: @489-491: "After hearing the two utterances, participants (all right-handed) were instructed to answer whether the second utterance sounded happy, sad, afraid, or neutral by pressing one of four keys (“S”, “D”, “F”, and “G”) with their left fourth, third, second, and first finger, respectively". This task may have been rather confusing and/or somewhat difficult for the participants: they had to memorize the association of the emotions happy, sad, afraid, or neutral, with 4 keyboard keys that had no meaning association with them, and had to click on the keys with 4 different fingers. Did the participants ever get confused? Did this impact the experiment in any way? Had some result to be excluded because of this? Please provide further details.

Response 2-2: We thank the reviewer for commenting on this point. The participants did not have to memorize the association between keys and emotion categories because this information was presented to them visually. We apologize for the lack of this description. We have added the following text to the Procedure subsection:

“After hearing the two utterances, four emotion labels with corresponding keys were presented visually (“[S] happy”, “[D] sad”, “[F] afraid”, “[G] identical”)”

In addition, we do not think that the responses of four different fingers influenced the results because the key positions were randomized across trials. This is described in the Procedure subsection:

“The correspondence of keys and response categories was randomized across trials.”

Comment 2-3: @497-499: why was there a difference in the presentation of terms in the 2 language groups (i.e., the Fr were given English terms and the JP were given JP terms)? The reason for this difference in the procedure needs to be explained. Also, details need to be provided to let the reader know whether this difference affected the results in any way.

Response 2-3: We agree with the reviewer that inconsistency in the presentation of terms could be problematic. We used English for the FR participants and Japanese terms for JP participants because we used the same presentation setting for the emotion recognition task in our previous study (Rachman et al. 2018). This study confirmed that both FR and JP participants can reliably recognize emotion categories with the current setting. In addition, the FR participants were given instructions on the definition of English emotion terms using the corresponding French terms. We thus do not think that the current experimental setting had an impact on the results.

Regarding the above point, we have added the following text in the Procedure subsection:

“Response categories in the recognition task for the French group in fact used the English terms (happy, sad, afraid) instead of the French equivalents, but were defined in the instructions using the equivalent French terms. We have confirmed in our previous study that emotion recognition tasks can be performed without problems in this presentation setting [22].”

Comment 2-4: Minor corrections in the text:

@line 280: 'in' is missing in: accuracy seen Experiment 1

@line 402: in 'lost in translation' the initial 'l' is not in italic

@line 479: The word recordings is repeated twice in: All sentence recordings recordings described...

Response 2-4: We thank the reviewer for pointing out our errors. We have corrected the following texts in the revised manuscript:

“the effect size of the native-language advantage in emotion recognition accuracy seen in Experiment 1”

“a lot more can expected to be lost in translation.”

“All sentence recordings described above (incl. reverse JP, reverse FR, and normal SE) were processed with DAVID”

---

## [Decision Letter · Decision Letter 1]

14 Apr 2023

Algorithmic voice transformations reveal the phonological basis of language-familiarity effects in cross-cultural emotion judgments

PONE-D-22-22873R1

Dear Dr. Nakai,

We’re pleased to inform you that your manuscript has been judged scientifically suitable for publication and will be formally accepted for publication once it meets all outstanding technical requirements.

Kind regards,

Federica Cavicchio

Academic Editor

PLOS ONE

Additional Editor Comments (optional):

Reviewers' comments:

Reviewer's Responses to Questions

**Comments to the Author**

1. If the authors have adequately addressed your comments raised in a previous round of review and you feel that this manuscript is now acceptable for publication, you may indicate that here to bypass the “Comments to the Author” section, enter your conflict of interest statement in the “Confidential to Editor” section, and submit your "Accept" recommendation.

Reviewer #2: All comments have been addressed

2. Is the manuscript technically sound, and do the data support the conclusions?

Reviewer #2: Yes

3. Has the statistical analysis been performed appropriately and rigorously? 

Reviewer #2: Yes

4. Have the authors made all data underlying the findings in their manuscript fully available?

Reviewer #2: Yes

5. Is the manuscript presented in an intelligible fashion and written in standard English?

Reviewer #2: Yes

6. Review Comments to the Author

Reviewer #2: My previous comments have been addressed. Thank you for the nice work and all best for your carreers

7. PLOS authors have the option to publish the peer review history of their article (what does this mean?). If published, this will include your full peer review and any attached files.

Reviewer #2: No

---

## [Editor Report · Acceptance letter]

24 Apr 2023

PONE-D-22-22873R1 

Algorithmic voice transformations reveal the phonological basis of language-familiarity effects in cross-cultural emotion judgments 

Dear Dr. Nakai:

I'm pleased to inform you that your manuscript has been deemed suitable for publication in PLOS ONE. Congratulations! Your manuscript is now with our production department. 

Kind regards, 

on behalf of

Dr. Federica Cavicchio 

Academic Editor

PLOS ONE